# Socioeconomic determinants of hypertension and prehypertension in Peru: Evidence from the Peruvian Demographic and Health Survey

Diego Chambergo-Michilot[1,2☯], Alexis Rebatta-Acuña[3], Carolina J. Delgado-Flores[4], Carlos J. Toro-Huamanchumo[5,6☯]*

1 Universidad Científica del Sur, Lima, Peru, 2 Red Latinoamericana de Cardiología, Lima, Peru, 3 Sociedad Científica de Estudiantes de Medicina de Ica, Universidad Nacional San Luis Gonzaga, Ica, Peru, 4 Universidad Nacional Jorge Basadre Grohmann, Tacna, Peru, 5 Universidad Católica Los Ángeles de Chimbote, Instituto de Investigación, Chimbote, Peru, 6 Clínica Avendaño, Unidad de Investigación Multidisciplinaria, Lima, Peru

☯ These authors contributed equally to this work.
* toro2993@hotmail.com

**Data Availability Statement:** The database is freely accessible from the website of the National

## Abstract

### Background

Peru is a Latin American country with a significant burden of hypertension that presents worrying rates of disparities in socioeconomic determinants. However, there is a lack of studies exploring the association between those determinants, hypertension and prehypertension in Peruvian population.

### Objective

We aimed to assess the association betwgeen socioeconomic determinants, hypertension and prehypertension using a nationally representative survey of Peruvians.

### Methods

We performed a cross-sectional analysis of the Peruvian Demographic and Health Survey (2018), which is a two-staged regional-level representative survey. We used data from 33,336 people aged 15 and older. The dependent variable was blood pressure classification (normal, prehypertension and hypertension) following the Seventh Report of the Joint National Committee (JNC-7) on hypertension management. Independent variables were socioeconomic: age, sex, marital status, wealth index, health insurance, education, region and area of residence. Due to the nature of the dependent variable (more than two categories), we opted to use the multinomial regression model, adjusting the effect of the multistage sample using the *svy* command. We tested interactions with the adjusted Wald test.

### Results

The prevalence of prehypertension and hypertension was 33.68% and 19.77%, respectively. Awareness was higher in urban than in rural areas (9.61% vs. 8.31%, p = 0.008). Factors

Institute of Statistics of Peru (http://iinei.inei.gob.
pe/microdatos/). The information can be obtained
by entering the survey query tab and selecting the
ENDES 2018 data using the modules modules #64,
#65 and #414.

**Funding:** The authors received no specific funding
for this work.

**Competing interests:** The authors have declared
that no competing interests exist.

associated with a higher prevalence ratio of both prehypertension and hypertension were age (ratios rose with each age group), male sex (prehypertension aRPR 5.15, 95%CI 4.63–5.73; hypertension aRPR 3.85, 95% CI 3.37–4.40) and abdominal obesity (prehypertension aRPR 2.11, 95%CI 1.92–2.31; hypertension aRPR 3.04, 95% CI 2.69–3.43). Factors with a lower prevalence ratio of both diseases were secondary education (prehypertension aRPR 0.76, 95%CI 0.60–0.95; hypertension aRPR 0.75, 95% CI 0.58–0.97), higher education (prehypertension aRPR 0.78, 95%CI 0.61–0.99; hypertension aRPR 0.62, 95% CI 0.46–0.82), being married/cohabiting (prehypertension aRPR 0.87, 95%CI 0.79–0.95; hypertension aRPR 0.77, 95% CI 0.68–0.87), richest wealth index (only prehypertension aRPR 0.76, 95% CI 0.63–0.92) and living in cities different to Lima (rest of the Coastline, Highlands and Jungle). Having health insurance (only hypertension aRPR 1.26, 95%CI 1.03–1.53) and current drinking (only prehypertension aRPR 1.15, 95%CI 1.01–1.32) became significant factors in rural areas.

## Conclusions

We evidenced socioeconomic disparities among people with hypertension and prehypertension. Better health policies on reducing the burden of risk factors are needed, besides, policy decision makers should focus on hypertension preventive strategies in Peru.

## Introduction

The morbidity of hypertension has increased by 18.6% in the general population in the last three decades [1]. According to the Global Burden of Disease, the percentage of years lived with disability for cardiovascular diseases increased by 34.3% in the 2007–2017 period [2]. Likewise, a study in 154 countries reported that the hypertension-related mortality was 106.3 deaths per 100,000 population in 2015 [1]. Considering the aforementioned scenario, hypertension represents a growing public health problem.

Whereas hypertension-related mortality has decreased in high-income countries, it has doubled in the low- and middle-income countries (LMICs) since 1990, reaching up to 85 deaths per 10,000 population in 2015 [1]. This trend has been attributed to socioeconomic disparities in the LMICs due to economic and demographic transitions. A systematic review revealed that the impact of the low income, lack of occupation and low educational level on the probability of hypertension was significantly higher compared to well-positioned levels in Latin America [3]. Previous studies have reported an association between some socioeconomic determinants, hypertension and prehypertension [4,5], nevertheless there are not studies exploring this association in Peru. Additionally, the LMICs are going through socioeconomic transitions that introduce challenges in the epidemiology of hypertension observing a notable disadvantage in groups with poor education and wealth. The impact of socioeconomic determinants on cardiovascular risk has become more evident with the Prospective Urban Rural Epidemiologic (PURE) study, which is an adult cohort in 20 countries [6]. This study found that the association between poor education and major cardiovascular disease was high and significant in the LMICs.

Peru is a Latin American LMIC with a significant burden of hypertension. A national report revealed that hypertension prevalence increased by 0.8 percentage points in the period 2017–2018 [7]. Moreover, the novel cut-off for diagnosis (ACC/AHA 2017, 130/80 mmHg) notably

increases the prevalence of hypertension in 18.5 percentage points in comparison with the traditional cut-off (JNC 7, 140/90 mmHg) [8]. This LMIC presents worrying rates of disparities in socioeconomic determinants. According to the Peruvian National Institute of Statistics and Informatics (INEI, Spanish initials), examples of those features are poor education, illiteracy and poverty, which are influenced by sex, ethnicity and area of residence [9,10]. These disparities are associated with hypertension; therefore, a comprehensive study of this relationship is needed.

Two relevant studies in the Peruvian population have been published. The PREVENCION study included 1,878 citizens from one province and explored the role of age, smoking, drinking, gender and diet [11]. The CRONICAS study, a prospective cohort in four sites of different urbanization and altitude in Peru, studied the association between hypertension and three socio-economic determinants (monthly family income, educational attainment, and assets index); however, researchers did not find any association [12]. These previous studies are limited by the lack of representativeness of the whole Peruvian population; moreover, researchers did not evaluate further socioeconomic features, such as type of residence or natural region. The assessment of socioeconomic determinants of high blood pressure is important as it identifies health gaps in order to implement novel public health prevention strategies in high-risk groups.

We aimed to 1) assess the association between socioeconomic determinants, hypertension and prehypertension using a nationally representative survey; and 2) explore the possible interactions between these socioeconomic determinants.

## Materials and methods

### Study design and data sources

We performed a cross-sectional analysis of the 2018 Peruvian Demographic and Family Health Survey (ENDES, Spanish initials) conducted by the INEI. The ENDES has nationwide, regional and urban-rural representativeness, and it is carried out yearly following the Demographic and Health Survey (DHS) Program guidelines. We used the information from the modules #64, #65 and #414. More detail about the ENDES methodology is available in the public technical report (http://iinei.inei.gob.pe/iinei/srienaho/Descarga/FichaTecnica/638-Ficha.pdf).

The sampling was two-staged, stratified, probabilistic, balanced and independent in urban and rural areas of each region. The sampling units were habitual residents of urban and rural households who spent the night before the survey in the selected household. Trained personnel visited each house and directly interviewed the individuals.

We included people aged 15 years or more, and excluded the individuals with missing and biologically non-plausible data in the independent and dependent variables.

### Study variables

**Dependent variable.** The dependent variable was blood pressure (BP) classification (normal, prehypertension and hypertension). Trained personnel measured blood pressure (BP) using an OMRON Automatic Upper Arm Digital Blood Pressure Monitor (HEM-7113 model). Individuals had to be seated with the right arm resting on a flat surface at the level of the heart; after five minutes at rest, the first measurement was taken, and two minutes later, the second measurement [7]. Blood pressure measurement ranged from 0 to 299 mmHg with an accuracy of ±3 mmHg, besides the monitor was calibrated. A standard arm (220–320 mm) or thicker arm (320–420 mm) cuff was used accordingly to the individual.

If an individual's 1) mean SBP ranged from 120 mmHg to 139 mmHg or mean DBP ranged from 80 mmHg to 89 mmHg; and 2) did not report hypertension history or taking anti-hypertensive medication; he/she was included in the prehypertension group. If an individual's 1) mean

SBP or DBP measurements were ≥140 mmHg or ≥90 mmHg, respectively; or 2) reported hypertension history (awareness) or taking anti-hypertensive medication, he/she was included in the hypertension group. This classification was according to The Seventh Report of the Joint National Committee on Prevention, Detection, Evaluation, and Treatment of High Blood Pressure [13].

**Independent variables: Socioeconomic determinants.** Independent variables were age (15–25, 26–35, 36–45, 46–55, 56–65 or >65 years), sex (male or female), marital status (not married/cohabiting or married/cohabiting), wealth index, having health insurance (yes or no), educational level (no education, primary, secondary or higher), region [Lima (Peru's capital), which is part of the Coastline, rest of the Coastline, Highlands or Jungle] and area of residence (urban or rural). The wealth index was calculated based on the methodology of Rutstein and Johnson [14,15]. This index is based on the characteristics of households and the availability of consumer durables directly related to the socioeconomic level. Each individual received a score generated for their household, and was, accordingly, assigned a quintile (poorest, poorer, middle, richer or richest).

**Other variables.** We assessed the following covariables: current smoker (cigarette consumption in the last month), current drinker (alcohol consumption in the last month) and abdominal obesity (yes or no). We used the recommended cut-off of abdominal obesity for Latin American populations (94 cm for men and 88 cm for women) [16].

## Data analysis

We used the Stata v.14.0 (Stata Corporation, College Station, Texas, USA) to analyze the data. The characteristics of the sample design and weighting factors were specified using the *svy* and *subpop* commands.

We described all the variables using absolute frequencies and weighted proportions with their 95% confidence intervals (95%CI). We used the Rao-Scott Chi-square test to assess the statistically significant differences between categorical variables (independent variables vs. outcome). We considered a p-value<0.05 as statistically significant.

Due to the nature of the dependent variable (more than two categories), we opted to use the multinomial regression model to identify the association between socioeconomic determinants, hypertension and prehypertension. We reported the association measures using the relative prevalence ratios (RPRs) and their 95%CI. Independent variables with a significant p-value and variables known to predict the dependent variable by the literature were included in the multivariate model. We assessed possible collinearity relationships among variables using the variance inflation factors. Finally, we tested possible interactions with the adjusted Wald test.

## Ethical considerations

ENDES 2018 is freely available at the INEI webpage (http://iinei.inei.gob.pe/microdatos/). We did not consider any information that could reveal individuals' identities. Moreover, potential participants had to give their consent to participate, and this survey did not involve biological samples and participants were free not to answer any question they felt uncomfortable with.

## Results

### Population characteristics

In 2018, 35,388 people were surveyed by the ENDES. According to our selection criteria, we excluded 2,052 individuals for several reasons (**Fig 1**). We analyzed the data of 33,336 participants.

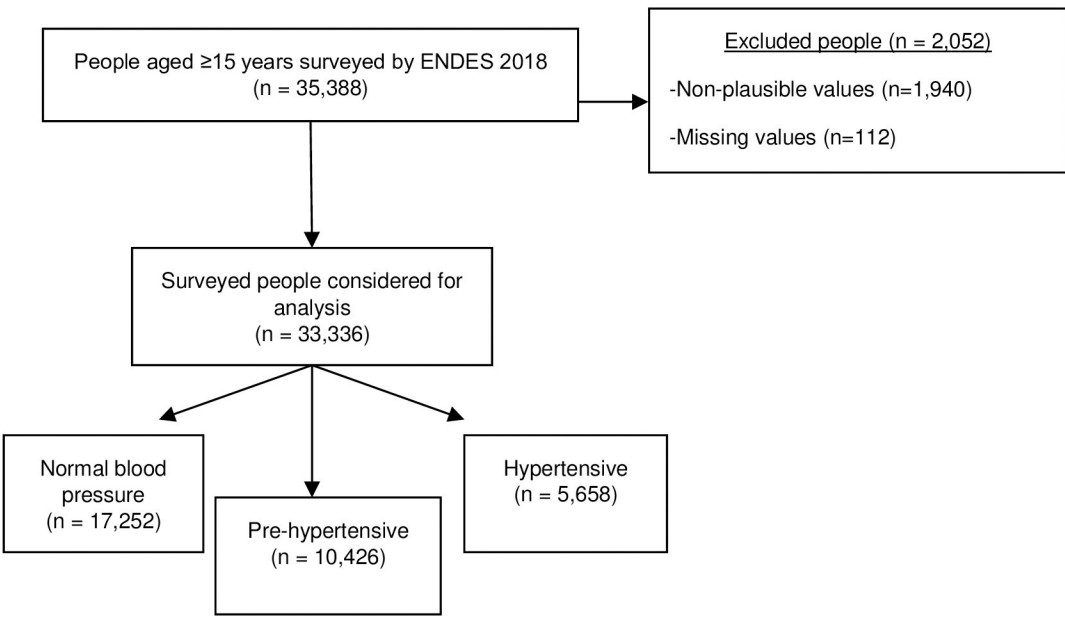

**Fig 1. Flowchart of the selection of study participants, ENDES 2018.**

The proportion of older adults was 10.99%. Proportions of female and male sex were similar, as well as the proportion of each wealth index quintile. Complete secondary education was the most frequent educational level (44.05%) and most people were living in urban areas (80.08%). The prevalence of prehypertension and hypertension was 33.68% and 19.77%, respectively (**Table 1**).

The hypertension awareness was significantly higher in urban areas (urban: 9.61%, rural: 8.31%, p = 0.008). Additionally, despite antihypertensive medication was also higher in urban areas, the differences were not significant (urban: 89.71%, rural: 85.75%, p = 0.121) (**S1 and S2 Figs**).

## Description of characteristics by BP classification

More than half of older adults were hypertensive (54.84%). Approximately half of the males were classified in the prehypertension group (45.88%). Prevalence of hypertension increased as the wealth index increased (percentage point difference (PPD) of hypertension between the richest and the poorest was +5.94), but decreased with educational level (PPD: -22.57). Prevalence of prehypertension increased with educational level (PPD between higher and "no education": +6.6). Both hypertension and prehypertension were higher among urban residents (vs. rural residents) and participants with abdominal obesity (vs. non abdominal obesity). All variables showed statistical significance (**Table 2**).

## Multinomial logistic regression analysis

In the crude analysis, the following factors: older age, male sex, living in urban areas and abdominal obesity significantly increased the prevalence ratio of both hypertension and prehypertension. Living in any region different from Lima significantly decreased the prevalence ratio of both diseases (**Table 3**).

The factors associated with a higher prevalence ratio of prehypertension in the multivariate model were older age, male sex and abdominal obesity. Factors associated with a lower

**Table 1. Characteristics of the population, ENDES 2018, n = 33336.**

| Characteristic | n | Weighted %[a] | 95% CI[a] |
|---|---|---|---|
| **Age** | | | |
| 15–25 | 7093 | 23.90 | 23.12–24.40 |
| 26–35 | 9272 | 20.88 | 20.24–21.54 |
| 36–45 | 6956 | 18.84 | 18.22–19.48 |
| 46–55 | 3899 | 15.09 | 14.41–15.78 |
| 56–65 | 3019 | 10.31 | 9.75–10.89 |
| >65 | 3097 | 10.99 | 10.42–11.58 |
| **Sex** | | | |
| Female | 18863 | 50.78 | 49.95–51.52 |
| Male | 14473 | 49.22 | 48.38–50.05 |
| **Marital status** | | | |
| Not married/cohabiting | 11066 | 38.22 | 37.37–39.07 |
| Married/cohabiting | 22270 | 61.78 | 60.93–62.63 |
| **Wealth index** | | | |
| Poorest | 10698 | 18.98 | 18.41–19.56 |
| Poorer | 8225 | 20.67 | 19.91–21.46 |
| Middle | 6173 | 20.90 | 20.18–21.64 |
| Rich | 4732 | 20.15 | 19.37–20.95 |
| Richest | 3508 | 19.30 | 18.47–20.16 |
| **Having health insurance** | | | |
| No | 19349 | 53.74 | 52.86–54.62 |
| Yes | 13987 | 46.26 | 45.38–47.14 |
| **Educational level** | | | |
| No education | 1683 | 3.87 | 3.59–4.18 |
| Primary | 8064 | 19.77 | 19.18–20.39 |
| Secundary | 14289 | 44.05 | 43.18–44.91 |
| Higher | 9300 | 32.31 | 31.46–33.17 |
| **Region** | | | |
| Lima | 3867 | 36.41 | 35.55–34.28 |
| Rest of the Coastline | 9527 | 25.73 | 24.88–26.60 |
| Highlands | 12241 | 25.50 | 24.55–26.48 |
| Jungle | 7701 | 12.36 | 11.75–12.99 |
| **Area of residence** | | | |
| Urban | 21771 | 80.08 | 79.59–80.57 |
| Rural | 11565 | 19.92 | 19.43–20.41 |
| **Abdominal obesity** | | | |
| No | 16558 | 47.28 | 46.36–48.19 |
| Yes | 16778 | 52.72 | 51.81–53.53 |
| **Current smoker** | | | |
| No | 29997 | 88.70 | 88.09–89.29 |
| Yes | 3339 | 11.30 | 10.71–11.91 |
| **Current drinker** | | | |
| No | 22849 | 65.06 | 64.16–65.94 |
| Yes | 10487 | 34.94 | 34.06–35.84 |
| **Blood pressure classification** | | | |
| Normal | 17252 | 46.54 | 45.66–47.43 |
| Prehypertension | 10426 | 33.68 | 32.91–34.46 |

(*Continued*)

**Table 1.** (Continued)

| Characteristic | n | Weighted %[a] | 95% CI[a] |
|---|---|---|---|
| Hypertension | 5658 | 19.77 | 19.06–20.50 |

[a] Confidence intervals and estimated proportions were calculated considering sampling specifications.

prevalence ratio of prehypertension were belonging to the richest group, secondary/higher education, being married/cohabiting and living in any region different to Lima.

Similarly, the factors associated with a higher prevalence ratio of hypertension in the multivariate model were older age, male sex, living in an urban area and abdominal obesity. Factors associated with a lower prevalence ratio of hypertension were secondary/higher education, being married/cohabiting and living in the Highlands.

## Stratified multinomial logistic regression analysis

We found interactions between the area of residence and the rest of the variables. In comparison to our non-stratified multivariate model, in urban areas, the prevalence ratio of age and male sex increased, but the ratio of educational levels became non-significant (**Table 4**).

In comparison to our non-stratified multivariate model, in rural areas, the prevalence ratio of age, male sex and abdominal obesity decreased. The ratio of region became non-significant, and the ratio of marital status increased. Having health insurance became a significant factor (only for hypertension) as well as current drinking (only for prehypertension) (**Table 5**).

## Discussion

### Main findings

We analyzed a nationally representative survey in Peru. Age and male sex increased the prevalence ratio of both hypertension and prehypertension, but education and living in the rest of Coastline and Highlands decreased it. Living in urban areas only increased the prevalence ratio of hypertension, and belonging to the highest wealth index group decreased the ratio of prehypertension. Area of residence modified the effect of the majority of socioeconomic determinants.

### Factors that increased the prevalence of hypertension and prehypertension

In the present study, the likelihood of having hypertension and prehypertension rises with increasing age. Cross-sectional and longitudinal studies in Iran and Portugal [5,17] have reported similar results. This association is biologically supported since aging produces arterial changes, such as stiffening of the arteries, endothelial dysfunction, collagen deposition in the ventricle walls, among others [18]. We consider necessary the implementation of primary health care strategies for the prevention of hypertension and related-cardiovascular-diseases, mainly among older adults [19].

Men were more likely to have hypertension and prehypertension compared to women. Similar results have been reported in previous studies. For example, Mahadir-Naidu *et al.* published a secondary data analysis of a national survey in Malaysia and found a significant association between male sex, hypertension and prehypertension [20]. However, a cohort study in Portugal reported that the incidence rate of hypertension increased with aging until 60 years in both genders. In older adults, the incidence rate stabilized in men and continued to rise in women [17]. The biological risk could be higher in men than in women, until menopause

**Table 2. Blood pressure category according to the socioeconomic determinants, ENDES 2018.**

| Socioeconomic determinants | Blood pressure classification (weighted %)[a] | | | P-value[b] |
|---|---|---|---|---|
| | Normal BP | Prehypertension | Hypertension | |
| **Age** | | | | |
| 15–25 | 63.64 | 31.77 | 4.61 | <0.001 |
| 26–35 | 56.36 | 34.57 | 9.09 | |
| 36–45 | 50.38 | 5.44 | 14.19 | |
| 46–55 | 36.17 | 37.23 | 26.70 | |
| 56–65 | 27.36 | 33.19 | 39.47 | |
| >65 | 16.42 | 28.76 | 54.84 | |
| **Sex** | | | | |
| Female | 60.23 | 21.87 | 17.89 | <0.001 |
| Male | 32.43 | 45.88 | 21.71 | |
| **Marital status** | | | | |
| Not married/cohabiting | 47.94 | 32.23 | 19.84 | 0.024 |
| Married/cohabiting | 45.69 | 34.59 | 19.73 | |
| **Wealth index** | | | | |
| Poorest | 49.94 | 33.14 | 16.94 | <0.001 |
| Poorer | 48.29 | 33.23 | 18.51 | |
| Middle | 46.81 | 34.17 | 19.03 | |
| Rich | 43.32 | 35.14 | 21.57 | |
| Richest | 44.44 | 32.69 | 22.88 | |
| **Having health insurance** | | | | |
| No | 58.28 | 28.44 | 13.28 | <0.001 |
| Yes | 32.95 | 39.79 | 27.31 | |
| **Educational level** | | | | |
| No education | 33.36 | 28.67 | 38.54 | <0.001 |
| Primary | 39.14 | 31.95 | 28.95 | |
| Secundary | 49.43 | 33.74 | 16.84 | |
| Higher | 48.77 | 35.27 | 15.97 | |
| **Region** | | | | |
| Lima | 42.87 | 36.14 | 20.99 | <0.001 |
| Rest of the Coastline | 46.66 | 32.46 | 20.89 | |
| Highlands | 49.54 | 32.73 | 17.74 | |
| Jungle | 50.97 | 30.94 | 18.08 | |
| **Area of residence** | | | | |
| Urban | 45.75 | 33.89 | 20.44 | <0.001 |
| Rural | 49.84 | 32.94 | 17.26 | |
| **Abdominal obesity** | | | | |
| No | 56.14 | 32.20 | 11.68 | <0.001 |
| Yes | 37.95 | 35.03 | 27.03 | |
| **Current smoker** | | | | |
| No | 47.42 | 32.7 | 19.90 | <0.001 |
| Yes | 39.62 | 41.51 | 18.89 | |
| **Current drinker** | | | | |
| No | 47.75 | 31.98 | 20.28 | <0.001 |
| Yes | 44.35 | 36.87 | 18.83 | |

[a] Confidence intervals and estimated proportions were calculated considering sampling specifications.

[b] P-values were calculated using the Rao-Scott Chi-square.

**Table 3. Association between socioeconomic determinants and hypertension status, ENDES 2018.**

| Socioeconomic determinants | Prehypertension | | | | Hypertension | | | |
|---|---|---|---|---|---|---|---|---|
| | cRPR | 95% CI | aRPR[a] | 95% CI | cRPR | 95% CI | aRPR[a] | 95% CI |
| **Age** | | | | | | | | |
| 15–25 | Ref. | | Ref. | | Ref. | | Ref. | |
| 26–35 | 1.23*** | 1.10–1.37 | 1.14* | 1.01–1.29 | 2.23*** | 1.79–2.76 | 1.96*** | 1.55–2.48 |
| 36–45 | 1.41*** | 1.26–1.58 | 1.25** | 1.10–1.42 | 3.89*** | 3.13–4.83 | 3.09*** | 2.44–3.92 |
| 46–55 | 2.06*** | 1.79–2.38 | 1.89*** | 1.60–2.22 | 10.16*** | 8.15–12.67 | 7.93*** | 6.23–10.09 |
| 56–65 | 2.43*** | 2.08–2.84 | 2.28*** | 1.90–2.73 | 19.94*** | 15.96–24.90 | 15.7*** | 12.17–20.25 |
| >65 | 3.51*** | 2.93–4.19 | 3.44*** | 2.76–4.29 | 46.16*** | 36.56–58.27 | 38.58*** | 29.30–50.80 |
| **Sex** | | | | | | | | |
| Female | Ref. | | Ref. | | Ref. | | Ref. | |
| Male | 3.9*** | 3.62–4.20 | 5.15*** | 4.63–5.73 | 2.25*** | 2.05–2.48 | 3.85*** | 3.37–4.40 |
| **Marital status** | | | | | | | | |
| Not married/cohabiting | Ref. | | Ref. | | Ref. | | Ref. | |
| Married/cohabiting | 1.13** | 1.04–1.22 | 0.87** | 0.79–0.95 | 1.04 | 0.95–1.15 | 0.77*** | 0.68–0.87 |
| **Wealth index** | | | | | | | | |
| Poorest | Ref. | | Ref. | | Ref. | | Ref. | |
| Poorer | 1.04 | 0.94–1.14 | 0.93 | 0.82–1.05 | 1.13* | 1.00–1.28 | 1.15 | 0.98–1.36 |
| Middle | 1.1 | 1.00–1.22 | 0.92 | 0.79–1.06 | 1.2** | 1.05–1.37 | 1.13 | 0.92–1.39 |
| Rich | 1.22*** | 1.10–1.37 | 0.9 | 0.77–1.07 | 1.47*** | 1.27–1.69 | 1.16 | 0.92–1.45 |
| Richest | 1.11 | 0.97–1.25 | 0.76** | 0.63–0.92 | 1.52*** | 1.29–1.78 | 1.03 | 0.80–1.31 |
| **Having health insurance** | | | | | | | | |
| No | Ref. | | Ref. | | Ref. | | Ref. | |
| Yes | 2.48*** | 2.29–2.67 | 1.01 | 0.91–1.12 | 3.64*** | 3.30–4.02 | 1.02 | 0.89–1.18 |
| **Educational level** | | | | | | | | |
| No education | Ref. | | Ref. | | Ref. | | Ref. | |
| Primary | 0.95 | 0.79–1.14 | 0.85 | 0.69–1.05 | 0.65*** | 0.54–0.78 | 0.87 | 0.70–1.09 |
| Secundary | 0.79* | 0.66–0.95 | 0.76* | 0.60–0.95 | 0.3*** | 0.25–0.36 | 0.75* | 0.58–0.97 |
| Higher | 0.84 | 0.70–1.01 | 0.78* | 0.61–0.99 | 0.29*** | 0.24–0.35 | 0.62** | 0.46–0.82 |
| **Region** | | | | | | | | |
| Lima | Ref. | | Ref. | | Ref. | | Ref. | |
| Rest of the Coastline | 0.83*** | 0.74–0.92 | 0.78*** | 0.69–0.88 | 0.92 | 0.80–1.05 | 0.86 | 0.74–1.01 |
| Highlands | 0.78*** | 0.71–0.87 | 0.76*** | 0.67–0.87 | 0.73*** | 0.64–0.84 | 0.77** | 0.65–0.91 |
| Jungle | 0.72*** | 0.65–0.80 | 0.69*** | 0.60–0.79 | 0.73*** | 0.63–0.83 | 0.86 | 0.72–1.02 |
| **Area of residence** | | | | | | | | |
| Rural | Ref. | | Ref. | | Ref. | | Ref. | |
| Urban | 1.12** | 1.04–1.21 | 1.09 | 0.97–1.22 | 1.29*** | 1.17–1.42 | 1.2* | 1.03–1.39 |
| **Abdominal obesity** | | | | | | | | |
| No | Ref. | | Ref. | | Ref. | | Ref. | |
| Yes | 1.61*** | 1.49–1.74 | 2.11*** | 1.92–2.31 | 3.43*** | 3.10–3.78 | 3.04*** | 2.69–3.43 |
| **Current smoker** | | | | | | | | |
| No | Ref. | | Ref. | | Ref. | | Ref. | |
| Yes | 1.52*** | 1.34–1.72 | 0.99 | 0.86–1.15 | 1.14 | 0.97–1.33 | 1.05 | 0.86–1.27 |
| **Current drinker** | | | | | | | | |
| No | Ref. | | Ref. | | Ref. | | Ref. | |
| Yes | 1.24*** | 1.15–1.35 | 1.01 | 0.92–1.10 | 1 | 0.91–1.11 | 1.06 | 0.95–1.20 |

cRPR: crude relative prevalence ratio. aRPR: adjusted relative prevalence ratio.

[a] We adjusted for age, sex, marital status, wealth index, having health insurance, education, region, area, obesity, being a current smoker and being a current drinker.

*p <0.05.

**p<0.01.

***p<0.001.

**Table 4. Association between socioeconomic determinants and hypertension status in urban areas, ENDES 2018.**

| Socioeconomic determinants | Prehypertension | | | | Hypertension | | | |
|---|---|---|---|---|---|---|---|---|
| | cRPR | 95% CI | aRPR[a] | 95% CI | cRPR | 95% CI | aRPR[a] | 95% CI |
| **Age** | | | | | | | | |
| 15–25 | Ref. | | Ref. | | Ref. | | Ref. | |
| 26–35 | 1.23** | 1.09–1.41 | 1.13 | 0.98–1.30 | 2.29*** | 1.78–2.95 | 2.03*** | 1.55–2.65 |
| 36–45 | 1.39*** | 1.21–1.59 | 1.19* | 1.02–1.39 | 4*** | 3.10–5.16 | 3.16*** | 2.42–4.14 |
| 46–55 | 2.21*** | 1.85–2.63 | 2.01*** | 1.66–2.44 | 11.11*** | 8.55–14.41 | 8.82*** | 6.63–11.74 |
| 56–65 | 2.68*** | 2.21–3.27 | 2.56*** | 2.01–3.26 | 24.41*** | 18.65–31.94 | 19.84*** | 14.56–27.02 |
| >65 | 4.04*** | 3.18–5.13 | 3.91*** | 2.92–5.23 | 57.67*** | 43.11–77.15 | 47.47*** | 33.73–66.82 |
| **Sex** | | | | | | | | |
| Female | Ref. | | Ref. | | Ref. | | Ref. | |
| Male | 4.15*** | 3.79–4.54 | 5.54*** | 1.92–2.40 | 2.43*** | 2.18–2.72 | 4.28*** | 3.64–5.03 |
| **Marital status** | | | | | | | | |
| Not married/cohabiting | Ref. | | Ref. | | Ref. | | Ref. | |
| Married/cohabiting | 1.2*** | 1.09–1.31 | 0.9 | 0.81–1.01 | 1.12 | 1.00–1.25 | 0.79** | 0.68–0.92 |
| **Wealth index** | | | | | | | | |
| Poorest | Ref. | | Ref. | | Ref. | | Ref. | |
| Poorer | 1.05 | 0.89–1.25 | 0.92 | 0.76–1.11 | 1.2 | 0.96–1.49 | 1.19 | 0.92–1.53 |
| Middle | 1.1 | 0.92–1.30 | 0.89 | 0.73–1.08 | 1.24 | 0.99–1.55 | 1.14 | 0.88–1.48 |
| Rich | 1.22* | 1.02–1.45 | 0.87 | 0.70–1.07 | 1.52*** | 1.21–1.90 | 1.14 | 0.87–1.51 |
| Richest | 1.1 | 0.92–1.33 | 0.73** | 0.58–0.92 | 1.57*** | 1.24–1.98 | 1.02 | 0.75–1.38 |
| **Having health insurance** | | | | | | | | |
| No | Ref. | | Ref. | | Ref. | | Ref. | |
| Yes | 3.04*** | 2.85–3.24 | 1 | 0.88–1.14 | 4.55*** | 4.21–4.92 | 0.97 | 0.82–1.14 |
| **Educational level** | | | | | | | | |
| No education | Ref. | | Ref. | | Ref. | | Ref. | |
| Primary | 0.84 | 0.58–1.21 | 0.87 | 0.59–1.30 | 0.7* | 0.50–0.98 | 1.08 | 0.72–1.63 |
| Secundary | 0.65* | 0.45–0.93 | 0.77 | 0.52–1.14 | 0.25*** | 0.18–0.35 | 0.91 | 0.60–1.38 |
| Higher | 0.66 | 0.47–0.94 | 0.8 | 0.54–1.20 | 0.21*** | 0.15–0.30 | 0.75 | 0.49–1.16 |
| **Region** | | | | | | | | |
| Lima | Ref. | | Ref. | | Ref. | | Ref. | |
| Rest of the Coastline | 0.82*** | 0.74–0.91 | 0.76*** | 0.68–0.86 | 0.94 | 0.82–1.08 | 0.86 | 0.74–1.00 |
| Highlands | 0.82*** | 0.73–0.92 | 0.81** | 0.71–0.92 | 0.76*** | 0.66–0.88 | 0.79* | 0.67–0.95 |
| Jungle | 0.66*** | 0.59–0.74 | 0.6*** | 0.52–0.69 | 0.74*** | 0.64–0.86 | 0.76** | 0.63–0.91 |
| **Abdominal obesity** | | | | | | | | |
| No | Ref. | | Ref. | | Ref. | | Ref. | |
| Yes | 1.73*** | 1.58–1.89 | 2.15*** | 1.92–2.40 | 3.85*** | 3.39–4.37 | 2.99*** | 2.57–3.48 |
| **Current smoker** | | | | | | | | |
| No | Ref. | | Ref. | | Ref. | | Ref. | |
| Yes | 1.46*** | 1.27–1.68 | 0.98 | 0.83–1.15 | 1.11 | 0.93–1.33 | 1.07 | 0.86–1.33 |
| **Current drinker** | | | | | | | | |
| No | Ref. | | Ref. | | Ref. | | Ref. | |
| Yes | 1.18 | 1.07–1.29 | 0.99 | 0.88–1.10 | 0.95 | 0.85–1.07 | 1.06 | 0.92–1.22 |

cRPR: crude relative prevalence ratio. aRPR: adjusted relative prevalence ratio.

[a] We adjusted for age, sex, marital status, wealth index, having health insurance, education, region, obesity, being a current smoker and being a current drinker.

*p <0.05.

**p<0.01.

***p<0.001.

**Table 5. Association between socioeconomic determinants and hypertension status in rural areas, ENDES 2018.**

| Socioeconomic determinants | Prehypertension | | | | Hypertension | | | |
|---|---|---|---|---|---|---|---|---|
| | cRPR | 95% CI | aRPR | 95% CI | cRPR | 95% CI | aRPR | 95% CI |
| **Age** | | | | | | | | |
| 15–25 | Ref. | | Ref. | | Ref. | | Ref. | |
| 26–35 | 1.17* | 1.01–1.36 | 1.22* | 1.04–1.44 | 1.9*** | 1.36–2.66 | 1.74* | 1.22–2.48 |
| 36–45 | 1.51*** | 1.28–1.78 | 1.51*** | 1.26–1.81 | 3.4*** | 2.44–4.76 | 2.78*** | 1.94–4.00 |
| 46–55 | 1.63*** | 1.34–1.98 | 1.57*** | 1.25–1.97 | 7.25*** | 5.14–10.24 | 5.42*** | 3.72–7.91 |
| 56–65 | 1.9*** | 1.55–2.34 | 1.77*** | 1.39–2.27 | 10.05*** | 7.19–14.05 | 6.92*** | 4.66–10.25 |
| >65 | 2.67*** | 2.15–3.32 | 2.59*** | 1.98–3.39 | 26.84*** | 18.97–37.99 | 19.64*** | 12.89–29.90 |
| **Sex** | | | | | | | | |
| Female | Ref. | | Ref. | | Ref. | | Ref. | |
| Male | 3.1*** | 2.78–3.47 | 3.95*** | 3.40–4.59 | 1.68*** | 1.45–1.95 | 2.68*** | 2.18–3.30 |
| **Marital status** | | | | | | | | |
| Not married/cohabiting | Ref. | | Ref. | | Ref. | | Ref. | |
| Married/cohabiting | 0.9 | 0.80–1.00 | 0.71*** | 0.62–0.81 | 0.82** | 0.71–0.95 | 0.69*** | 0.57–0.83 |
| **Wealth index** | | | | | | | | |
| Poorest | Ref. | | Ref. | | Ref. | | Ref. | |
| Poorer | 0.95 | 0.81–1.09 | 0.87 | 0.75–1.02 | 1 | 0.83–1.20 | 1.09 | 0.88–1.34 |
| Middle | 1.04 | 0.76–1.41 | 0.93 | 0.67–1.29 | 1.22 | 0.80–1.87 | 1.18 | 0.74–1.89 |
| Rich | 1.32 | 0.87–2.01 | 1.29 | 0.82–2.03 | 1.4 | 0.80–2.46 | 1.43 | 0.79–2.61 |
| Richest | 1.32 | 0.60–2.90 | 0.94 | 039–0.24 | 1.39 | 0.60–3.18 | 1.49 | 0.54–4.12 |
| **Having health insurance** | | | | | | | | |
| No | Ref. | | Ref. | | Ref. | | Ref. | |
| Yes | 2.37*** | 2.18–2.58 | 0.99 | 0.85–1.14 | 3.63*** | 3.25–4.05 | 1.26* | 1.03–1.53 |
| **Educational level** | | | | | | | | |
| No education | Ref. | | Ref. | | Ref. | | Ref. | |
| Primary | 0.97 | 0.80–1.18 | 0.81 | 0.66–1.01 | 0.45*** | 0.37–0.55 | 0.7** | 0.55–0.90 |
| Secundary | 0.78* | 0.64–0.85 | 0.7** | 0.55–0.91 | 0.18*** | 0.14–0.22 | 0.57*** | 0.42–0.77 |
| Higher | 0.97 | 0.74–1.25 | 0.83* | 0.53–0.99 | 0.3*** | 0.21–0.41 | 0.56** | 0.37–0.84 |
| **Region** | | | | | | | | |
| Lima | Ref. | | Ref. | | Ref. | | Ref. | |
| Rest of the Coastline[†] | - | - | - | - | - | - | - | - |
| Highlands | 0.86 | 0.72–1.04 | 0.87 | 0.71–1.06 | 0.93 | 0.75–1.16 | 0.91 | 0.69–1.20 |
| Jungle | 0.94 | 0.78–1.15 | 0.98 | 0.79–1.21 | 0.93 | 0.72–1.19 | 1.23 | 0.91–1.65 |
| **Abdominal obesity** | | | | | | | | |
| No | Ref. | | Ref. | | Ref. | | Ref. | |
| Yes | 1.17** | 1.04–1.30 | 1.83*** | 1.62–2.07 | 2.18*** | 1.90–2.50 | 2.94*** | 2.52–3.42 |
| **Current smoker** | | | | | | | | |
| No | Ref. | | Ref. | | Ref. | | Ref. | |
| Yes | 1.82*** | 1.52–2.18 | 1.1 | 0.90–1.35 | 1.14 | 0.89–1.45 | 0.95 | 0.71–2.27 |
| **Current drinker** | | | | | | | | |
| No | Ref. | | Ref. | | Ref. | | Ref. | |
| Yes | 1.54*** | 1.37–1.75 | 1.15* | 1.01–1.32 | 1.12 | 0.95–1.31 | 1.17 | 0.97–1.41 |

cRPR: crude relative prevalence ratio. aRPR: adjusted relative prevalence ratio.

[a] We adjusted for age, sex, marital status, wealth index, having health insurance, education, region, obesity, being a current smoker and being a current drinker.

[†]Not considered because this category had too many strata omitted because they contained no subpopulation members.

*p <0.05

**p<0.01

***p<0.001

onset when women begin to experience several blood pressure changes, which are promoted by sympathetic activity and oxidative stress-like response to estradiol decrease [21]. Differences between our study and the Portuguese cohort might be caused by population characteristics. In this cohort, more than 60% were women, and the majority of adults were aged 40 years of more, which is the common menopause onset. It partially explain the rise of hypertension incidence in women since menopause is a risk factor. It is also important to mention that hypertensive Peruvian men could be more exposed to harmful lifestyles that could trigger high blood pressure, such as depression status [22] and alcohol consumption [23].

Previous representative studies from Bangladesh [24] and India [25] have found an association between living in urban areas and hypertension, which was consistent with our results. In contrast, in Malaysia [20], rural areas were associated with a higher prevalence of hypertension. This association mainly depends on the social setting and definition of urbanization. Peru is a middle-income country that has gone through a period of internal migration in the last century [26], and has not yet reached the absolute decentralization in health care. This situation could favor that most of diagnoses and treatments were made in urban areas, promoting migration and increasing the number of cases, as we reported. Despite this, the PERU MIGRANT study evaluated the effect of migration on the incidence of hypertension and reported a higher risk of hypertension in rural non-migrants compared with rural-urban migrants and urban non-migrants. However, this study was conducted only in two cities, and each one presents its own sociodemographic characteristics and migration profile, limiting the external validity to the whole Peruvian population [27]. To standardize, some authors have proposed new definitions of urbanization using complex scales [28]. For example, a district-level analysis in India showed a significant positive association between urbanization scale and hypertension [29]. It is advisable to generate new evidence on urban disparities using complex definitions in Peru, especially since extrapolating information from other countries may be wrong due to the variability of social settings.

## Factors that decreased the prevalence of prehypertension and hypertension

Similar to previous studies [20], we found that in the general population, a secondary/higher education was associated with a lower prevalence of hypertension and prehypertension. However, studies carried out in Bangladesh disagreed with our results [24,30]. These differences may be attributable to variation of educational access among countries. It is understood that people with a higher educational level are better informed about cardiovascular risk factors and consequently will have healthy lifestyles. In 2019, a cross-sectional study from 204 Latin American communities reported that people with higher educational attainment have a higher likelihood of controlled hypertension [31]. According to our results, education could be an excellent way to prevent hypertension and prehypertension.

We found that living in different cities (excluding Lima) was associated with a lower prevalence of both hypertension (for highlands) and prehypertension. There are approximately 400 healthcare centers in Lima due to high population density, therefore hypertension diagnoses rate could be higher than in other locations of the country, which frequently are less populated. After controlling for common socioeconomic characteristics and harmful habits, the association was statistically significant; thus other explanations should be explored. For example, it has been suggested that air pollution particles may have a relevant role in BP changes [32].

## Interaction with the area of residence

Among rural residents, marital status, having health insurance and current drinking had a significant role in prehypertension and hypertension prevalence. Mental disorders' effect on

hypertension is a well-known fact [22], and these disorders often affect subjects in rural areas [33]. Promotion of a healthy and unstressful life by marital status may be an explanation for its protective role in hypertension [34], and the higher prevalence of married couples in the rural setting could support our results [35]. Regarding health insurance, our findings were supported by studies that reported that people with health insurance are more likely to fo for screening exams for chronic diseases, including hypertension [36]. We found an urban-rural gap in hypertension awareness that resulted in an independent association between health insurance and hypertension in rural areas, but not urban. This highlights the importance of having a decentralized healthcare system that guarantees early detection of the most prevalent non-communicable diseases in Peru, such as hypertension. Finally, alcohol-related vascular and cardiac changes may explain the association between current drinking and hypertension [37]. Our results suggest that current drinking is associated with prehypertension, but only in rural areas. Differences in patterns of alcohol consumption, which not only involves the frequency but the type of beverage, may explain the differences [38].

Among urban residents, belonging to the richest group was associated with a lower prevalence of prehypertension. Results of a Peruvian national survey revealed that approximately 60% of the surveyed people in the highest socioeconomic status had a daily consumption of fruits and vegetables [39]. This habit could reduce the probability of early BP changes due to lesser stressful lifestyle and a healthier diet.

## Relevance for public health

We analyze the association between several socioeconomic determinants for hypertension and prehypertension, and we resume our results in three key messages. First, early blood pressure screening and timely treatment of groups in risk, such as rural residents living without a couple or with alcohol problems, could reduce hypertension incidence. Second, the urban-rural gap of hypertension awareness highlights the need for more efforts to provide health insurance for citizens and decentralize the Peruvian health system. Third, education could lead to the adoption of healthy lifestyles, which are essential to prevent cardiovascular diseases. Policymakers and primary healthcare providers should consider this for health campaigns.

Future studies should focus on new interventions for high cardiovascular risk groups (e.g., Adults with hypertension). While the Peruvian Health Ministry approved a hypertension guideline in 2015, there have been no recommendations on actions to improve primary prevention. Additionally, there is no national plan in Peru to establish preventive measures. This is different from Argentina, where authorities proposed the empowerment of surveillance of the hypertensive population, dissemination of guidelines, training and promotion of sodium reduction policies [40]. This plan was developed in collaboration with the HEARTS initiative, which is a novel plan to support governments in primary healthcare prevention of cardiovascular diseases.

## Limitations

Some limitations should be highlighted. First, the cross-sectional study design does not allow us to assess causality between our study variables. Second, the self-report variables could introduce a recall bias in our data interpretation; however, the dependent variable was defined using more confident methods (automatic tensiometer). Third, although interviewers were previously trained, they could make mistakes; nevertheless, ENDES follows the DHS model, so there is a methodological strength in data collection. Fourth, since this was a secondary data analysis, some confounding variables, such as time of hypertension diagnosis, stress or anxiety, could not be assessed.

## Conclusion

Prehypertension and hypertension are frequent diseases among the Peruvian population. While aging, male sex and urban area increased the prevalence of hypertension, education and living in the rest of Coastline and Highlands decreased it. Besides, the area of residence was an important effect modifier. Peru needs more health policies focused on cardiovascular risk groups. Policymakers should start to lay the foundations for a preventive plan in Peru, considering the different socioeconomic characteristics of the population.

## Supporting information

**S1 Fig. Awareness of hypertension by area of residence, ENDES 2018.**
(DOCX)

**S2 Fig. Antihypertensive medication (among hypertensive adults) by area of residence, ENDES 2018.**
(DOCX)

## Acknowledgments

Diego Chambergo-Michilot thanks Mrs. Maria C. for her emotional support and recommendations for the study.

## Author Contributions

**Conceptualization:** Diego Chambergo-Michilot, Carlos J. Toro-Huamanchumo.

**Data curation:** Diego Chambergo-Michilot, Carolina J. Delgado-Flores, Carlos J. Toro-Huamanchumo.

**Formal analysis:** Diego Chambergo-Michilot, Carlos J. Toro-Huamanchumo.

**Investigation:** Diego Chambergo-Michilot, Carlos J. Toro-Huamanchumo.

**Methodology:** Diego Chambergo-Michilot, Carlos J. Toro-Huamanchumo.

**Validation:** Diego Chambergo-Michilot, Carlos J. Toro-Huamanchumo.

**Visualization:** Diego Chambergo-Michilot, Carlos J. Toro-Huamanchumo.

**Writing – original draft:** Diego Chambergo-Michilot, Alexis Rebatta-Acuña, Carolina J. Delgado-Flores, Carlos J. Toro-Huamanchumo.

**Writing – review & editing:** Diego Chambergo-Michilot, Alexis Rebatta-Acuña, Carolina J. Delgado-Flores, Carlos J. Toro-Huamanchumo.

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
