## [Decision Letter · Decision Letter 0]

29 Dec 2020

PONE-D-20-30393

Socioeconomic determinants of hypertension and prehypertension in Peru: evidence from the Peruvian Demographic and Health Survey

PLOS ONE

Dear Dr. Toro-Huamanchumo,

Thank you for submitting your manuscript to PLOS ONE. After careful consideration, we feel that it has merit but does not fully meet PLOS ONE’s publication criteria as it currently stands. Therefore, we invite you to submit a revised version of the manuscript that addresses the points raised during the review process.

We look forward to receiving your revised manuscript.

Kind regards,

Antonio Palazón-Bru, PhD

Academic Editor

PLOS ONE

Journal Requirements:

We note that one or more of the authors are employed by a commercial company: Tau-RELAPED Group.

2.1. Please provide an amended Funding Statement declaring this commercial affiliation, as well as a statement regarding the Role of Funders in your study. If the funding organization did not play a role in the study design, data collection and analysis, decision to publish, or preparation of the manuscript and only provided financial support in the form of authors' salaries and/or research materials, please review your statements relating to the author contributions, and ensure you have specifically and accurately indicated the role(s) that these authors had in your study. You can update author roles in the Author Contributions section of the online submission form.

2.2. Please also provide an updated Competing Interests Statement declaring this commercial affiliation along with any other relevant declarations relating to employment, consultancy, patents, products in development, or marketed products, etc.  

2.3. Please know it is PLOS ONE policy for corresponding authors to declare, on behalf of all authors, all potential competing interests for the purposes of transparency. PLOS defines a competing interest as anything that interferes with, or could reasonably be perceived as interfering with, the full and objective presentation, peer review, editorial decision-making, or publication of research or non-research articles submitted to one of the journals. Competing interests can be financial or non-financial, professional, or personal. Competing interests can arise in relationship to an organization or another person. Please follow this link to our website for more details on competing interests: http://journals.plos.org/plosone/s/competing-interests

Reviewers' comments:

Reviewer's Responses to Questions

**Comments to the Author**

1. Is the manuscript technically sound, and do the data support the conclusions?

Reviewer #1: Yes

2. Has the statistical analysis been performed appropriately and rigorously? 

Reviewer #1: Yes

3. Have the authors made all data underlying the findings in their manuscript fully available?

Reviewer #1: Yes

4. Is the manuscript presented in an intelligible fashion and written in standard English?

Reviewer #1: Yes

5. Review Comments to the Author

Reviewer #1: Thank you for selecting me to review the paper. The study covers an important topic which is nicely written and used nationally representative data. Some minor addition would be helpful to improve the paper. My suggestions are below:

Abstract

1. Check again the format of Plos one (I think it requires a background paragraph in the abstract.

2. Line 33: abbreviate JNC-7

3. Lines 35-37, can add a line regarding justification why you have used multinomial regression but no need to put STATA code in the abstract

4. Better to add ratios with Cis in abstract to make it understandable to readers just looking at abstract of the study

Introduction

1. Check reference style of plosone

2. Line 77: Is there any report on HTN prior 2017?

3. At the end of the introduction, should incorporate why this study is important in general and for the country. Specifying ‘only two studies available with limited sample’ does not properly justify the importance of the study

Data analysis

Line 154: database or data?

Line 161: A justification would be better why authors have used multinomial regression models (dependent variable is nominal…)

Table 1: add total ‘n’ in the title of the table, add comma in specifying “n”

Age 15-25, report double decimal places (weighted %)

Wealth index: Middle, Richest: report double decimal places (weighted %) and check this for rest of tables

Line 195: ‘not’ significant instead of ‘no’.

Table 2: Title, try another word instead of ‘hypertension status’

Discussion

Lines 282-83, same country?

6. PLOS authors have the option to publish the peer review history of their article (what does this mean?). If published, this will include your full peer review and any attached files.

Reviewer #1: No

---

## [Author Response · Author response to Decision Letter 0]

1 Jan 2021

Comment: We note that one or more of the authors are employed by a commercial company: Tau-RELAPED Group.

Response: Thank you for your observation. Although this is not a commercial company, we decided to remove this affiliation. 

Comment: Check again the format of Plos one (I think it requires a background paragraph in the abstract).

Response: Thank you for your observation. We added the following sentence in background paragraph in the Abstract: “Peru is a Latin American country with a significant burden of hypertension that presents worrying rates of disparities in socioeconomic determinants. However, there is a lack of studies exploring the association between those determinants, hypertension and prehypertension in Peruvian population”.

Comment: Please provide an amended Funding Statement declaring this commercial affiliation, as well as a statement regarding the Role of Funders in your study. If the funding organization did not play a role in the study design, data collection and analysis, decision to publish, or preparation of the manuscript and only provided financial support in the form of authors' salaries and/or research materials, please review your statements relating to the author contributions, and ensure you have specifically and accurately indicated the role(s) that these authors had in your study. You can update author roles in the Author Contributions section of the online submission form.

Response: The present study was self-funded. This was stated in the manuscript.

Comment: Please also provide an updated Competing Interests Statement declaring this commercial affiliation along with any other relevant declarations relating to employment, consultancy, patents, products in development, or marketed products, etc.

Response: Thank you for your observation. We added a Competing Interest Statement as follows: “Any author have a conflict of interest. Any affiliation does not alter our adherence to PLOS ONE policies on sharing data and materials”.

Comment: Line 33: abbreviate JNC-7 

Response: Thank you for your observation. We replaced JNC-7 with “Seventh Report of the Joint National Committee (JNC-7) on hypertension management”.

Comment: Lines 35-37, can add a line regarding justification why you have used multinomial regression but no need to put STATA code in the abstract

Response: Thank you for your observation. We added the following line: “Due to the nature of the dependent variable (more than two categories), we opted to use the multinomial regression model”.

Comment: Better to add ratios with Cis in abstract to make it understandable to readers just looking at abstract of the study

Response: Thank you for your observation. We added the ratios and confidence intervals.

Comment: Check reference style of plosone.

Response: Thank you for your observation. We added the reference style of PLOS ONE in the references.

Comment: Line 77: Is there any report on HTN prior 2017?

Response: Yes, there is. There are annual reports of only hypertension prevalence. The last report was published in 2018, and we reported it in the paper. 

Comment: At the end of the introduction, should incorporate why this study is important in general and for the country. Specifying ‘only two studies available with limited sample’ does not properly justify the importance of the study.

Response: Thank you for your observation. We added the following line in the end of the introduction: “The assessment of socioeconomic determinants of high blood pressure is important as it identifies health gaps in order to implement novel public health prevention strategies in high-risk groups”.

Comment: Line 154: database or data?

Response: Thank you for your observation. We replaced “database” with “data”.

Comment: Line 161: A justification would be better why authors have used multinomial regression models (dependent variable is nominal…)

Response: Thank you for your observation. We added the following line: “Due to the nature of the dependent variable (more than two categories), we opted to use the multinomial regression model to identify the association between socioeconomic determinants, hypertension and prehypertension”.

Comment: Table 1: add total ‘n’ in the title of the table, add comma in specifying “n”

Response: Thank you for your observation. We added the total “n” in the title of the Table 1: “, n = 33336”

Comment: Age 15-25, report double decimal places (weighted %)

Response: Thank you for your observation. We added the correct number (23.90).

Comment: Wealth index: Middle, Richest: report double decimal places (weighted %) and check this for rest of tables

Response: Thank you for your observation. We added the correct numbers in all tables.

Comment: Line 195: ‘not’ significant instead of ‘no’.

Response: Thank you for your observation. We have changed the word “no” to “not” in the line 195.

Comment: Table 2: Title, try another word instead of ‘hypertension status’

Response: Thank you for your observation. We have changed the word “hypertension status” to “blood pressure category”

Comment: Lines 282-83, same country?

Response: Thank you for your observation. We added the following line: “Cross-sectional and longitudinal studies in Iran and Portugal (5,17) have reported similar results”.

---

## [Decision Letter · Decision Letter 1]

7 Jan 2021

Socioeconomic determinants of hypertension and prehypertension in Peru: evidence from the Peruvian Demographic and Health Survey

PONE-D-20-30393R1

Dear Dr. Toro-Huamanchumo,

We’re pleased to inform you that your manuscript has been judged scientifically suitable for publication and will be formally accepted for publication once it meets all outstanding technical requirements.

Kind regards,

Antonio Palazón-Bru, PhD

Academic Editor

PLOS ONE

Additional Editor Comments (optional):

Reviewers' comments:

Reviewer's Responses to Questions

**Comments to the Author**

1. If the authors have adequately addressed your comments raised in a previous round of review and you feel that this manuscript is now acceptable for publication, you may indicate that here to bypass the “Comments to the Author” section, enter your conflict of interest statement in the “Confidential to Editor” section, and submit your "Accept" recommendation.

Reviewer #1: All comments have been addressed

2. Is the manuscript technically sound, and do the data support the conclusions?

Reviewer #1: Yes

3. Has the statistical analysis been performed appropriately and rigorously? 

Reviewer #1: Yes

4. Have the authors made all data underlying the findings in their manuscript fully available?

Reviewer #1: Yes

5. Is the manuscript presented in an intelligible fashion and written in standard English?

Reviewer #1: Yes

6. Review Comments to the Author

Reviewer #1: All comments have been addressed by authors. The paper has now improved much and can be accepted to publish in the journal

7. PLOS authors have the option to publish the peer review history of their article (what does this mean?). If published, this will include your full peer review and any attached files.

Reviewer #1: No

---

## [Editor Report · Acceptance letter]

18 Jan 2021

PONE-D-20-30393R1 

Socioeconomic determinants of hypertension and prehypertension in Peru: evidence from the Peruvian Demographic and Health Survey 

Dear Dr. Toro-Huamanchumo:

I'm pleased to inform you that your manuscript has been deemed suitable for publication in PLOS ONE. Congratulations! Your manuscript is now with our production department. 

Kind regards, 

on behalf of

Dr. Antonio Palazón-Bru 

Academic Editor

PLOS ONE